# Are Multiple Chemosensory Systems Accountable for COVID-19 Outcome?

**DOI:** 10.3390/jcm10235601

**Published:** 2021-11-28

**Authors:** Antonio Caretta, Carla Mucignat-Caretta

**Affiliations:** 1Department of Food and Drug Science, University of Parma, 43100 Parma, Italy; antonio.caretta@unipr.it; 2NIBB—National Institute for Biostructures and Biosystems, 00136 Rome, Italy; 3Department of Molecular Medicine, University of Padova, 35131 Padova, Italy

**Keywords:** SARS-CoV-2, COVID-19, olfaction, taste, carotid body, oxygen sensing

## Abstract

Chemosensory systems (olfaction, taste, trigeminus nerve, solitary chemoreceptor cells, neuroendocrine pulmonary cells, and carotid body, etc.) detect molecules outside or inside our body and may share common molecular markers. In addition to the impairment of taste and olfaction, the detection of the internal chemical environment may also be incapacitated by COVID-19. If this is the case, different consequences can be expected. (1) In some patients, hypoxia does not trigger distressing dyspnea (“silent” hypoxia): Long-term follow-up may determine whether silent hypoxia is related to malfunctioning of carotid body chemoreceptors. Moreover, taste/olfaction and oxygen chemoreceptors may be hit simultaneously: Testing olfaction, taste, and oxygen chemoreceptor functions in the early stages of COVID-19 allows one to unravel their connections and trace the recovery path. (2) Solitary chemosensory cells are also involved in the regulation of the innate mucosal immune response: If these cells are affected in some COVID-19 patients, the mucosal innate immune response would be dysregulated, opening one up to massive infection, thus explaining why COVID-19 has lethal consequences in some patients. Similar to taste and olfaction, oxygen chemosensory function can be easily tested with a non-invasive procedure in humans, while functional tests for solitary chemosensory or pulmonary neuroendocrine cells are not available, and autoptic investigation is required to ascertain their involvement.

## 1. Introduction

In order to survive, our body monitors the external and internal environment by detecting molecules through specialized receptors. The perception of external molecules mainly occurs through olfaction, taste, and trigeminal chemesthesis, which is the chemical sense mediated by the trigeminal nerve. These chemical senses have gained popularity for their involvement in interactions with the external environment during many daily activities, from food search/consumption to mate recognition, and started to gain popularity with the 2004 Nobel prize awarded to Buck and Axel for the discovery of olfactory receptors [1]. However, chemical signaling inside our body is also crucial for survival: Internal chemical sensing relies on receptors for oxygen, carbon dioxide, pH (hydrogen ions), plus several other molecules, such as glucose, fatty acids, and mediators of inflammation among others, that are present at many sites. These receptors are always active to monitor biochemical properties of the “milieu interieur” for maintaining homeostasis balance.

Traditionally, chemoreception was investigated by completely separated fields of research, pertaining to the perception of external (e.g., taste, olfaction) or sensing of internal chemical stimuli (e.g., respiratory gases, pH). However, it is apparent that a clear-cut distinction between conscious and unconscious perception is not tenable, since each chemosensory system is not exclusively tuned to selected molecules, as was thought until a few decades ago. In actuality, chemosensory mechanisms hosted in different organs may be considered polymodal sensory systems that share some sensing molecules and transduction pathways. Olfactory neurons bear olfactory receptor proteins, yet they also express other receptors, for example, trace amine-associated receptors (TAARS) [2] and many receptors for hormones, to a large extent involved in metabolic regulation [3,4,5], but not limited to this function [6]. On the other hand, olfactory receptors are “ectopically” expressed at many sites [7] and contribute to various functions: In the pancreas, they modulate insulin release [8] and one olfactory receptor is involved in glucose metabolism [9]. Taste receptor cells, in addition to taste stimuli, are activated by inflammation mediators [10] and are sensitive to hormones that control metabolism, contributing to the regulation of food intake [4]. Similarly, chemosensory cells sensitive to internal signals are not narrowly tuned as previously alleged. For example, carotid body cells are sensitive to blood gases but also to glucose [11] and inflammatory mediators [12,13]; solitary chemoreceptor cells recognize microbial chemosignals [14] but also express sweet taste receptors that modulate the antimicrobial response [15]; pulmonary neuroendocrine cells can sense oxygen, carbon dioxide, and other different molecules, and are involved in ventilatory function and immunomodulation [16,17,18]; in the gastrointestinal tract, solitary chemoreceptor cells (aka tuft cells) recognize different microbial signals [19] while enterochromaffin cells also recognize glucose among other molecules [20].

All these chemosensory systems share common fundamental molecular markers: Carotid body cells, enterochromaffin cells, and pulmonary neuroendocrine cells express canonical olfactory receptors [21,22,23] and solitary chemosensory cells express canonical taste receptors [14]. Moreover, all the afferent signals of these chemosensory systems are clearly separated in the periphery, but in the central nervous system, they surprisingly converge on the same brain areas of the brainstem, subcortical, and cortical areas.

It is thus reasonable to assume that these chemosensory systems may also share a common sensitivity to the SARS-CoV-2 virus, given their similar molecular machinery. This idea is also supported by the fact that their function may be altered in COVID-19.

COVID-19 has been extensively studied since the beginning of the pandemic. A puzzling symptom of COVID-19 is the frequent impairment and even loss of taste and olfaction, reported by patients and/or clinically tested [24,25,26], that may resolve within days or weeks, but in some cases persists after months [27]. These chemosensory deficits are not specific to COVID-19 and have also been reported for other diseases, including viral infections [28], neurodevelopmental [29] and neurodegenerative disorders [30], cancer independently from treatments [31], metabolic diseases [32], and hypertension [33].

In the face of life-threatening pneumonia, taste/olfaction involvement is usually considered a minor problem by caregivers. In clinics, this inference may be presumably correct, because by themselves these symptoms are not lethal and, in most patients, they are transient [34]. Recently, attention has been paid to olfactory and taste systems in the light of the so-called LONG-COVID, the long-term consequences of COVID-19 [35], as the potential entry site of the virus to the brain and/or as a potential predictor of long-term brain damage or functional impairment.

The phenomenon named silent hypoxia refers to the lack of respiratory response (called dyspnea) to a life-threatening decrease in oxygen saturation of hemoglobin [36]. Several COVID-19 patients, either post-pneumonia or not, display silent hypoxia, since they fail to report any distress when in short of oxygen, for example after a brief physical exercise [36]. A major player in detecting oxygen levels in the blood is the carotid body, whose chemosensory function was postulated nearly a century ago [37]. In addition to the carotid body, an oxygen-sensitive potassium channel is present in neuroepithelial bodies/neuroendocrine cells in the lung, which may help in monitoring oxygen levels [38]. In COVID-19 patients, immune responses appear also deregulated. The solitary chemosensory cells (SCC) are a different class of chemoreceptors, known as gut tuft cells or airway brush cells [39,40]. Interestingly, they may also play a role in triggering immune responses [41]. Hence, it would be interesting to explore the role of different chemoreceptors in eliciting some yet unexplained COVID-19 symptoms.

## 2. A Hypothesis on Chemosensory Involvement in COVID-19

It is suggested that, in addition to taste, olfaction, and trigeminal chemesthesis, other chemoreceptor cells (for example, carotid body chemoreceptors and solitary chemoreceptors cells) may be affected by COVID-19, with potentially dire consequences for patients. This hypothesis links the different chemosensory systems as possible targets of SARS-CoV-2. It gives rise to separate predictions that can be tested in the field, as detailed below.

### 2.1. Testing the Hypothesis

#### 2.1.1. Silent Hypoxia as a Test Case

When COVID-19 results in acute respiratory distress syndrome (ARDS), some patients may experience the so-called “Silent Hypoxia”: They show a life-threatening drop in hemoglobin oxygen saturation, despite being unaware of their oxygen shortage, that usually would trigger severe dyspnea [42]. These patients are deprived of a crucial alerting mechanism that drives the ventilatory responses and conscious behavior to counteract dangerous hypoxia. Its deficiency causes a delay in seeking proper treatment, with possible negative outcomes for patients’ health. This failure may persist in ARDS survivors [43].

Several explanations for silent hypoxia mechanisms have been proposed:
The lack of hypoxic pulmonary vasoconstriction (HPV) [44,45].A possible inflammation of the central nervous system at various levels and with different mechanisms [46,47,48,49].The dysregulation of the renin-angiotensin system [50].It has been proposed that the oxygen chemoreceptor function may have been impaired [51,52].Lastly, it has also been suggested that COVID-19 silent hypoxia is not surprising and may be referred to as normal neurophysiological mechanisms [36,53].


No evidence is available to rule out or accept these different hypotheses, but the simplest explanation resides in the impairment of oxygen chemoreceptor cell function, mainly but not exclusively hosted in carotid bodies. Failure of these oxygen chemoreceptors could explain the absence of dyspnea as well as the absence of hyperpnea or polypnea. Moreover, the failure of pulmonary oxygen chemoreceptor cells/neuroepithelial bodies might explain the absence of hypoxic pulmonary vasoconstriction.

The possibility that damage to respiratory gas sensing and of taste/olfaction/trigeminal chemesthesis is tightly related is suggested by observations highlighting some challenging analogies between them:
Anatomically, all of them host sustentacular cells, crucial for receptor cell survival and key entry sites for SARS-CoV-2. Moreover, stem cells are present in each organ, allowing cell turnover [54], which may drive the path to recovery.Afferents from the carotid body and from the posterior third of the tongue run together in the glossopharyngeal nerve through the petrosal ganglion to reach the solitary tract nucleus [55], targeting partially overlapping areas in the brainstem.Functionally, in taste buds, there are receptors sensitive to pH (sour taste receptors) that act similarly to ectopic chemoreceptors present in the larynx [56] and, most interestingly, in the carotid body [57].Receptors for carbon dioxide are also present in the olfactory mucosa [58] and in the mouth [59].Both the olfactory bulb and the carotid body host a large number of dopaminergic cells [60].


Interestingly, canonical olfactory receptor proteins, typical of olfactory mucosa, are expressed “ectopically” in carotid body cells and may trigger the hypoxic ventilatory responses, contributing to the maintenance of oxygen balance by controlling breathing when oxygen levels fall, a mechanism that relies on the carotid body [23,61]. Moreover, canonical olfactory receptors are present in pulmonary neuroendocrine cells, a class of polymodal sensors, also involved in oxygen sensing [22].

In all these chemoreceptor systems, ACE2 receptors are present and warrant the entry of SARS-CoV2 [62,63]. Moreover, in one patient, SARS-CoV2 was detected in the carotid body at autopsy [64].

For these reasons, we suggest that the hypothesis stated above can be split in two testable predictions, which can be validated (see Table 1):
Silent hypoxia is related to the malfunctioning of chemosensory cells, mainly carotid body chemoreceptors.Taste and olfactory chemoreceptor cells are likely to be affected by the virus, together with carotid body chemoreceptors. Hence, taste and olfaction malfunction can be exploited as predictors of silent hypoxia. At present, no observation is available on the co-occurrence of these different chemosensory deficits, yet this hypothesis can be experimentally tested in a non-invasive way, as detailed in Table 1.


In the case of a person (Table 1, case 1) with normal taste/olfaction, normal hypoxic ventilatory response, and no silent hypoxia, nothing can be concluded about the abovementioned hypotheses. In the case of silent hypoxia with otherwise normal responses (case 2), it can be concluded that silent hypoxia does not depend on carotid body malfunctioning because of the presence of a normal hypoxic ventilatory response, while hypothesis II cannot be tested because taste, olfaction, and the carotid body are functioning correctly. Case 3 shows impaired carotid body functions with hypoxic ventilatory responses of which the patient is aware (no silent hypoxia), hence hypothesis I is falsified because the carotid body is not functioning while silent hypoxia is absent; also, hypothesis II is falsified because taste and olfaction are working, and carotid bodies are not. Case 4 has silent hypoxia and an impaired hypoxic ventilatory response, resulting in the verification of hypothesis I because carotid body impairment and silent hypoxia co-occur, while hypothesis II is falsified, because taste and olfaction are functioning while carotid bodies are not. Case 5 is a patient with only smell/taste disorders: In this case, hypothesis I cannot be tested because of the absence of both silent hypoxia and altered ventilatory response, while hypothesis II can be rejected because taste/olfaction are impaired while carotid bodies are functioning. If silent hypoxia is present together with smell/taste impairments but with a normal hypoxic ventilatory response (case 6), both hypotheses are falsified: Hypoxic response is present, hence carotid bodies are functioning, but silent hypoxia is present (hypothesis I), while taste/olfaction are impaired while the carotid body is functioning (hypothesis II). Case 7 is a patient with smell/taste disorders and altered hypoxic ventilatory responses, but without silent hypoxia: In this case, hypothesis I is falsified because of the absence of silent hypoxia, while hypothesis II is verified because of the co-occurrence of taste/smell disorders and carotid body impairments. In case 8, all the responses are abnormal, which verifies both hypotheses because of the impairment of carotid body functioning in the presence of silent hypoxia and taste/smell disorders.

**Table 1 jcm-10-05601-t001:** A synopsis of the possible findings in patients, at diagnosis.

	Taste/Olfaction Tests *	Hypoxic Ventilatory Response	Silent Hypoxia	Hypothesis I Silent Hypoxia Depends on Carotid Body	Hypothesis II Carotid Body and Taste/Olfaction Chemoreceptors Are Affected Together
1.	Normal	Normal	Absent	Not falsified, not verified	Not falsified, not verified
2.	Normal	Normal	**Present**	Falsified. Silent hypoxia not dependent on carotid body functioning	Not falsified, not verified
3.	Normal	**Not normal**	Absent	Falsified. Hypoxia dependent on carotid body functioning.	Falsified
4.	Normal	**Not normal**	Present	Verified	Falsified
5.	**Not normal**	Normal	Absent	Not falsified, not verified.	Falsified
6.	**Not normal**	Normal	Present	Falsified	Falsified
7.	**Not normal**	**Not normal**	Absent	Falsified	Verified
8.	**Not normal**	**Not normal**	Present	Verified	Verified

In bold: Pathological findings. * At least one not-normal value is sufficient for inclusion in the “not normal” group

#### 2.1.2. Involvement of Solitary Chemoreceptor Cells

Solitary chemosensory cells, also known as tuft cells in the gut or brush cells in airways, are a diffuse system of chemoreceptor cells. Although still commonly considered a marginal field of research both in physiology and immunology, in the last decade, new interesting data have been collected: These cells express the complex molecular machinery of taste chemoreceptor cells, and detect bitter and, to a lesser extent, sweet molecules [14,65,66]. Both on respiratory mucosa and intestinal mucosa they are known to trigger a mucosal innate immune response [67]. In addition, on respiratory mucosa, these chemosensory cells have been shown to contribute to breath control by activating afferent nervous fibers [68,69]. It is worth considering that persons that do not express taste receptor T2R38 have a worse prognosis if affected by COVID-19 [70], even if it is unclear whether this receptor is lacking in the tongue only or throughout the body. T2R38 is a bitter taste receptor initially found in the tongue, but also in the upper respiratory tract, where it may detect substances produced by Gram-negative bacteria, leading to the killing of bacteria and mucus clearance [71].

At present, no information is available on solitary chemosensory cells in COVID-19 patients. We suggest that the loss of function of solitary chemosensory cells may jeopardize the mucosal innate immune response, thus paving the way to a devastating entry of a large amount of virus particles. This may explain why the course of this disease is quite different among patients, ranging from asymptomatic infection to death.

The innate response mediated by solitary chemoreceptor cells is demonstrated in respiratory mucosa only for bacteria [14]. It is unknown whether solitary chemosensory cells express the ACE2 receptor, but it is not necessary that chemosensory cells are directly destroyed by the virus, since the loss of function can be achieved by the damage of nearby cells, similarly to SARS-CoV-2 action in the olfactory mucosa [72].

At present, the solitary chemoreceptor cell function cannot be clinically examined in patients, yet this system should be investigated post-mortem. The possibility of exploring this issue in animal models remains open.

Lastly, a diffuse impairment of chemosensory systems may also explain the worsening of metabolic regulation and the inflammatory response in already compromised, fragile patients.

## 3. Conclusions

We suggest that, while apparently different for structure, location, intracellular signaling, and sensitivity, the shared molecular signature of the different chemoreceptor systems may underlie common SARS-CoV-2 vulnerability.

From a clinical perspective, we propose that, starting in the early stages of infection on an outpatient basis, it may be worth measuring olfaction and taste functions, with simple tests or surveys [26,27,73,74,75] together with the oxygen chemoreceptor response using the Transient Test for Hypoxic Ventilatory Response [76,77] and six-minute walking test [78,79]. In this way, it would be possible to understand if the impairment of taste and olfaction correlates with carotid body chemoreceptor impairment and silent hypoxia (see Table 1) and if it is possible to predict which patients are at risk of silent hypoxia, notwithstanding the fact that the actual degree of involvement of each chemosensory system may differ in each person.

Lastly, we suggest that in COVID-19 patients, genetic or functional analysis of bitter taste receptors as well as, unfortunately under the present circumstances, systematic autoptic investigations of chemosensory systems, is warranted.

## Data Availability

Data are available in the cited literature.

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
