# Peer review of "Are Multiple Chemosensory Systems Accountable for COVID-19 Outcome?"

_jcm, 2021, doi:10.3390/jcm10235601_

Round 1
Reviewer 1 Report
well written and a good structure of the work, interesting the correlation to covid-olfaction-taste and chemosensory.A good study of the chemosensory and accurate analysis during COVID. It’s relevant to understand how is the effect of covid on chemosensory. It is intestìnig into the study of the chemosensory.
Author Response
We thank Reviewer 1 for the kind comment.
Reviewer 2 Report
I read with great interest the proposed work, which is well written and innovative.
I suggest to consider only to summarize table 1 in the text by proposing the different hypotheses grouped together on the basis of the answers in the first three columns
Author Response
We thank the Reviewer for the suggestion: we have added a narrative text summarizing the content of Table 1 (lines 166-191).Reviewer 3 Report
I find the hypotheses of the authors intriguing and timely as they aim at providing mechanistic insight on the shared involvement of the chemical senses. I also particularly like that the review - though perhaps more an opinion or a perspective - offers testable hypotheses for animal models and post-mortem autoptical work in humans.
There are, however, a few imprecisions that require correction before advising for publication:
(line 30)Chemesthesis should not be referred to as the trigeminal system only, as other systems are involved.
"largest audience with the 2004 Nobel prize...(line 33)" It is hard to quantify which one is the largest audience, and one may advance that COVID-19 provided the largest audience for the chemical senses. Please rephrase.
"pertaining to conscious perception of external... (line 42)" The conscious/unconscious dichotomy described here is incorrect. Smells and tastes, described as part of the external chemical stimuli, can also be perceived implicitly and unconsciously (e.g., priming). Please restate.
Please cite papers demonstrating that chemosensory deficits have been reported also for other diseases (line 78)
Author Response
REV. (line 30) Chemesthesis should not be referred to as the trigeminal system only, as other systems are involved.
AUTHORS: We thank the Reviewer for the thoughtful comment and apologize for our previous misleading phrasing. We completely agree with the reviewer, since chemesthesis is sense scattered through the body. We changed the text by specifying ‘trigeminal chemesthesis’ throughout the text (lines 30-31, 101, 132).
REV. "largest audience with the 2004 Nobel prize...(line 33)" It is hard to quantify which one is the largest audience, and one may advance that COVID-19 provided the largest audience for the chemical senses. Please rephrase.
AUTHORS: The Reviewer is right, now the text is changed accordingly (line 33).
REV. "pertaining to conscious perception of external... (line 42)" The conscious/unconscious dichotomy described here is incorrect. Smells and tastes, described as part of the external chemical stimuli, can also be perceived implicitly and unconsciously (e.g., priming). Please restate.
AUTHORS: We completely agree and thank the Reviewer for having noted this. Now the text is amended accordingly (lines 41-42).
REV. Please cite papers demonstrating that chemosensory deficits have been reported also for other diseases (line 78)
AUTHORS: Six papers have been added to Bibliography, showing the involvement of chemosensory deficits in other diseases, like viral infections, neurodevelopmental (autism, epilepsy, Tourette syndrome, 22q11 deletion, Fragile X etc.) and neurodegenerative (Parkinson and Alzheimer’s) disorders, cancer (independently from chemotherapy or radiotherapy), metabolic diseases (type 2 diabetes) and hypertension (lines 77-79).
This manuscript is a resubmission of an earlier submission. The following is a list of the peer review reports and author responses from that submission.
Round 1
Reviewer 1 Report
nice and interesting work. A good study of the chemosensory and accurate analysis during COVID. It’s relevant to understand how is the effect of covid on chemosensory. It is intestìnig into the study of the chemosensory.
Reviewer 2 Report
- I do not have a tool for detecting plagiarism, so I cannot say I am the best judge to assess for this.
- The sentence starting with line 86, is not a full sentence and does not clearly make the point the author is trying to say (i.e. that silent hypoxia may be a sign of sensory dysfunction due to SARS-COV-2)
- The paragraph starting with line 86, seem to be separate statement sentences that are almost unrelated or at least, not tied together in any way other than descriptively.
- Section 2.11 is mostly speculation, but it is in intriguing to think about.
- Line 174-175: "persons that do not express taste receptor T2R38 have a worse prognosis if affected by COVID-19 [64]. unless I look at the reference, I do not know what these receptors are.
- Line 177- but do we know that loss of these bitter taste receptors means they are absent elsewhere?